# Investigations of the Microstructure and Mechanical Properties of 17-4 PH ss Printed Using a MarkForged Metal X

**DOI:** 10.3390/ma15196898

**Published:** 2022-10-05

**Authors:** Adugna D. Akessa, Wakshum M. Tucho, Hirpa G. Lemu, Jørgen Grønsund

**Affiliations:** Faculty of Science and Technology, University of Stavanger, N-4036 Stavanger, Norway

**Keywords:** additive manufacturing, Markforged Metal X, mechanical properties, microstructure, heat treatment, stainless steel

## Abstract

The Markforged Metal X (MfMX) printing machine (Markforged Inc., Massachusetts, USA) is one of the latest introduced additive manufacturing (AM) devices. It is getting popular because of its safety, simplicity, and ability to utilize various types of powders/filaments for printing. Despite this, only a few papers have so far reported the various properties and performances of the components fabricated by the MfMX printer. In this study, the microstructure and mechanical properties of MfMX-fabricated 17-4 stainless steel (ss) in the as-printed and heat-treated conditions were investigated. XRD and microscopy analyses revealed a dominant martensitic microstructure with some retained austenite phase. The microstructure is generally characterized by patterned voids that were unfilled due to a lack of fusion between the adjacent filaments. Disregarding these defects (voids), the porosity of the dense region was less than 4%. Depending on the heat treatment conditions, the hardness and tensile strength were enhanced by 17–28% and 21–27%, respectively. However, the tensile strength analyzed in this work was low compared with some previous reports for L-PBF-fabricated 17-4 ss. In contrast, the hardness of the as-printed (331 ± 28 HV) and heat-treated samples under the H900 condition (417 ± 29 HV) were comparable with (and even better than) some reports in the literature, despite the low material density. The results generally indicated that the Markforged printer is a promising technology when the printing processes are fully developed and optimized.

## 1. Introduction

Additive manufacturing (AM) is a progressively developing manufacturing process that uses a 3D design model to fabricate products using diverse materials, including metals, ceramics, and plastics, for different applications [1]. AM is defined by ASTM F2792−12a as “processes of joining materials to make objects from 3D model data, usually layer upon layer, as opposed to subtractive manufacturing fabrication methodologies.” [2]. The technology was developed from simple applications that were intended for rapid prototyping and is now being used for the fabrication of functional products that can produce complex parts that are difficult to produce using traditional fabrication techniques [3].

Though the AM processes of metal printing have a complex non-equilibrium physical, chemical, and metallurgical nature depending on the process and material used, the part-fabrication techniques used in all AM technologies are mostly identical, where the final part is manufactured through the layer-by-layer addition of materials [4,5,6]. They use some form of energy source to selectively melt the material in metal powder form and a laser scans the regions in the powder layer, resulting in localized melting and solidification of the powder to form a layer of the part until the part building is completed [7]. In these processes, it is possible to produce a part that has an approximately 99% or higher density and has better properties than parts produced using the casting process [8]. However, due to several physical mechanisms involved, these methods are complex. The scanning speed is very high and phase changes occur over a very short time scale. All these complexities directly impact the process and influence the properties of the processed material. Moreover, the cost of these metal printers is not affordable for many companies and educational institutions [9,10].

The newly introduced Metal X system (Markforged Inc. USA), also referred to as atomic diffusion additive manufacturing (ADAM), is less complex, inexpensive, and has a larger printing volume capacity [11,12]. Like fused deposition modeling (FDM) technology, the Markforged Metal printing system uses materials in a filament form and deposits them layer-by-layer to create the desired part. This fabrication process is safe and simple with no need for powders, lasers, solvents, or volatile compounds. Besides the low cost of the equipment, the possibility of multi-material deposition and the availability of varieties of filament types for printing make this fabrication process popular. In FDM, the filament is extruded through the nozzle and deposited on the building platform layer upon layer based on the shape of the 3D CAD model and solidifies on the building platform, which is at a temperature below the melting temperature of the filament [13]. The Markforged Metal X system, on the other hand, uses filament-formed metal powder enclosed in a polymer, where the polymer reduces the toxicity and flammability risk of the filament. The metal powder used in the Metal X system is the same as that used in metal powder bed fusion, except that they are enclosed in the polymer [14].

Due to its good weldability [15] and high market demand, 17-4 HP (precipitation hardening) stainless steel is one of the several metals that was proposed by Markforged for industrialized production using the MfMX process [16]. The alloy 17-4 ss has a wide range of applications, such as aerospace, marine, nuclear, and chemical processes, because of its excellent mechanical strength and good corrosion resistance [15]. It is mainly strengthened by the precipitation of highly dispersed Cu-rich nanoparticles [17,18,19]. The hardening process involves a solid solution heat treatment at high temperatures and a subsequent aging heat treatment [18,20,21] that can lead to phase transformation and precipitation of the hardening components. The typical aging temperature for 17-4 ss is 480–620 °C [15]. The optimal tensile strength and hardness can be achieved under the H900 condition that adopts a solid solution heat treatment at 1038 °C for 0.5–1 h and aging treatment at 482 °C for 1 h. This is realized by the formation of Cu-rich precipitates (bcc) that maintain a coherent relationship with the matrix [20].

Since the Markforged Metal X 3D printing technology is a recently introduced AM process on the market, the parts that are fabricated by the machine have not been extensively studied, and consequently, only a few papers have been published in the literature so far. In this work, we fabricated 17-4 ss using a Markforged Metal X printer and studied the microstructure, hardness, and tensile strength of the resulting as-printed and H900 condition samples. The results from different testing and characterization techniques are presented and discussed herein.

## 2. Experimental

### 2.1. Materials and Methods

The stainless steel 17-4 samples studied in this work were manufactured using a Markforged Metal X 3D printing machine at the University of Stavanger, Norway. The average composition of 17-4 stainless steel is shown in Table 1.

The feedstock material for the MfMX printing system is a wire filament with a diameter of about 1.75 mm. The cross-sectional view of the filament and the magnified image that displays the details inside the filament (powder) are shown in Figure 1a and Figure 1b, respectively. The microstructure of the filament shows spherical powder particles of variable sizes ranging from a few nanometers to ca. 5 µm.

There are four basic steps in the MfMX process for manufacturing metal parts: design, printing, debinding, and sintering [22]. The design step is the conversion of a virtual model into printable layers using the slicer code “Eiger”, and transfers that information to the Metal X printer. This is followed by the selective extrusion of filaments bound with the binder (polymer) through a nozzle and printing layer by layer on the building plate. The first printed part at this step is called the “green part”. It is soft since it is bound in a polymer matrix. The “green part” is then debonded by immersing it in a specialized solvent that dissolves the primary binder thermally to form the “brown part”. The final step involves drying the “brown part” and sintering at a high temperature (about 85% of the melting temperature) in a horizontal tube furnace to transform the lightly bound “brown part” into a relatively dense metallic part. Sintering is done under argon and argon–hydrogen mixed media to prevent the parts from being contaminated. For the ease of removing the sintered component from the support, ceramic layers are deposited with a secondary extruder before starting the printing of the actual material. Detailed descriptions of the MfMX manufacturing process are available in [16,22,23].

Figure 2 illustrates successive layers that are fabricated with MfMX. The layers of printed parts have two tool paths (wall and infill), as shown in the figure. The “wall” is the tool path on the periphery, whereas the “infill” is the inner section of the part. The “wall” is oriented parallel with the edges of the part (parallel and normal to the part’s longer axis), while the “infill” is oriented ±45° with the part axis, as shown in the magnified sections of the sketches in Figure 2. The printing of the samples was done based on the “default parameters” setup proposed by the manufacturer [23]. Some of the known parameters applied for printing 17-4 ss with the MfMX system are given in Table 2.

### 2.2. Heat Treatment

The common practice in industries to improve the mechanical properties of the 17-4 stainless steel materials for various applications is a post-fabrication heat treatment. In this work, we adopted the industry standard, namely, an H900 heat treatment scheme, to attain the optimal mechanical strength. Accordingly, the samples were first solid solution heat-treated at 1038 °C for a 0.5 or 1 h duration. When the soaking time was over, the samples were either cooled in air or water (quenched). This step normally softens the material by releasing the initial residual stresses partially/completely. To increase the mechanical strength through the formation of precipitation hardening, each sample was aged at 482 °C for 1 h. As shown in [24], Cu-rich fine particles, which are coherent with the bcc matrix, are precipitated after the aging heat treatment. In addition, other samples were directly aged (DA) to analyze the effects of bypassing the homogenization treatment. All the heat treatments were done in a furnace (Nabertherm P300, Lilienthal/Bremen, Germany) equipped with a K-type thermocouple. To avoid undesirable phase transformations at lower temperatures, the samples were introduced after stabilizing the furnace at the given target temperature. The samples that were solid solution heat-treated and aged are referred to as H1A, H2A, H1Q, and H2Q. “H” stands for the two steps of heat treatment, whereas the numbers 1 and 2 refer to the soaking times of 0.5 and 1 h, respectively. Similarly, the letter “A” refers to air-cooling and “Q” stands for quenching. Furthermore, to examine the effects of the solution heat treatment alone, two samples were subjected to a solid solution heat treatment for 0.5 h and tested for hardness. These samples were S1A and S1Q, where “S” stand for the solution heat treatment (not aged). The list and the heat treatment descriptions of the samples are given in Table 3. The microstructure and mechanical properties of the heat-treated samples were analyzed and compared with those of the as-printed (AP) samples.

### 2.3. Hardness and Tensile Testing

Hardness tests were performed using a Vickers Struers DuraScan testing machine under a 5 kg HV force for a dwell time of 10 s. The surfaces of the specimens were finely polished according to the requirements for Vickers hardness testing. The hardness measurements were done on the surface normal to the building direction. The interval between adjacent indentations was 1 mm and the closest indentation near the edge was about 3 mm. The average hardness was calculated from 12 measurements for each of the specimens tested.

The tensile test was performed using an Instron 5985 universal tensile testing machine whose maximum loading capacity was 250 kN at a strain rate of 0.00007 s^−1^. The specimens for the tensile test were prepared following the ISO6892-1 standard preparation procedures [25]. To ensure production repeatability, six samples were fabricated by employing the same parameters (Table 2) for every condition shown in Table 3 for the tensile measurements. The schematic diagram of the specimens is shown in Figure 3.

### 2.4. Microstructure and Phases

The microstructure and fracture surfaces of the specimens were analyzed with scanning electron microscopy (SEM) using a Gemini SUPRA 35VP (ZEISS) equipped with EDAX energy dispersive X-ray spectroscopy (EDS). Sample preparation for the microstructure analysis consisted of mechanical grinding, fine polishing, and ultra-polishing with OP-S colloidal silica. Light optical microscopy (LOM, Olympus GX53) was also used for the microstructure and phase analysis. After the investigation with SEM, the samples were used for the measurement of Vickers hardness testing. Phase analysis was done using powder X-ray diffractograms recorded with Bruker D8 X-Ray diffraction (XRD) equipment with Cu_Kα_ radiation (λ = 1.54060 Å) and operating at 40 kV and 25 mA. The diffractograms were recorded within the 2θ range of 35° and 120° at a step size of 0.034°. Further investigations of the phases and microstructure were performed with transmission electron microscopy (TEM), JEOL-2100 (LaB_6_ filament) operating at 200 kV. For the TEM analysis, thin foils were prepared, first by thinning them down mechanically to a thickness of about 100 µm and then punching 3 mm disks from the thin foils. These disks were then electropolished using a dual jet polishing system, namely, Struers TENUPOL-5, operated at 13 V and −30 °C in an electrolyte solution of 95% methanol and 5% perchloric acid.

## 3. Results and Discussions

### 3.1. Phase Identification

XRD measurements were performed on all the samples to identify the phases before and after the heat treatment. The diffractograms recorded before (AP) and after the heat treatments (H1A, H2A, H1Q, H2Q, and DA) are shown in Figure 4. The peaks were indexed to α-bcc ferrite/martensite (a = 2.86 Å) and γ-fcc (a = 3.56 Å) phases. The intensities and the widths of the peaks varied, but all the samples contained both phases. The peaks that represented the fcc phase were only observed for low-index planes (111 and 200) and they were too weak compared to those of the bcc phase. From the intensities of the peaks, it could be inferred that the bcc phase was the dominant phase in the Markforged-Metal-X-manufactured 17-4 ss.

The TEM images also show the presence of martensitic structures in the as-printed MfMX-17-4 ss samples, as shown in Figure 5a,b. The insert in Figure 5a is a SAD pattern recorded from the same image. The SAD is indexed to a bcc structure in the [111] orientation. The bcc phase identified using both XRD and TEM was an α′ bcc martensite phase rather than a ferrite bcc phase. The latter is usually characterized by well-defined grain boundaries without laths. As we can see elsewhere, the SEM and LOM also confirmed martensite as the dominant phase in the MfMX-17-4 ss.

All the diffractograms (including for AP) demonstrated textured crystallography pertaining to the (110) planes of the bcc structure. This meant that the material was mainly martensitic even before the heat treatments. Factors that favor the retention of a large fraction of austenite in the AM 17-4 ss are suggested in the literature. These are (1) the presence of strain at high-angle grain boundaries, which suppresses the transformation of austenite to martensite; (2) a higher dislocation density; (3) supersaturation of the γ phase with the corresponding phase stabilizing elements; (4) smaller grain sizes; (5) interdendritic spacing; and (6) a powder manufacturing environment. The details can be referred to in [25].

As shown in the insert of Figure 4, it is evident that a considerable amount of austenite material was transformed into bcc/martensite material after the heat treatment, but not a very large quantity. Based on the intensities of the peaks, the volume fraction of the fcc phase appeared to decrease after the heat treatment (H2Q, H2A). Since the retained austenite material in the as-printed material was low, the transformed fraction was also low. However, a quantitative study is required to confirm this.

There are variations in the literature regarding the amount of martensite/austenite present in AM-manufactured 17-4 ss depending on the fabrication environment. Selective laser melting (SLM)-fabricated 17-4 ss, which was fabricated from the powder atomized in N_2_ for instance, mainly has an fcc structure (15% martensite) in a nitrogen gas environment fabrication [26]. However, fully martensitic 17-4 ss parts can be fabricated using SLM if the Ar-atomized powder is fabricated in a nitrogen environment according to Murr and co-workers [26]. On the other hand, the study by Cheruvathur et al. [17] indicated a 50% retained austenite and 50% bcc/martensite structure from SLM-built 17-4 ss using a nitrogen-atomized powder in a nitrogen environment. Two possible reasons are suggested by Murr et al. [26] for the high volume fraction of the retained austenite phase. The first reason is that nitrogen is considered to be a stabilizer of austenite, and thus, the absorption of the gas can lead to a reduction in martensitic transformation. The other reason stated by the authors was the formation of very fine subgrains that tend to shift the martensite transformation temperature. This suggests that careful choice and control of the gases utilized for the powder and the component fabrication of 17-4 ss using AM are important. In contrast to an AM-fabricated material, the as-received wrought 17-4 ss exhibits a fully martensitic/body-centered cubic structure [20,27]. A full martensite transformed as-built component is normally an advantage since it avoids post-processing, especially heat treatment at high temperatures.

Though MfMX and SLM techniques have notable differences, the effects of the cover gases in both cases must be identical since the thermodynamic transformation of the phases is likely similar. In the current study, the gas utilized in the sintering chamber of the MfMX printing system was Ar and a mixture of Ar and hydrogen. As can be inferred based on the XRD and microscopy analysis, the amount of austenite in the as-printed material observed in this work was significant. This signals that the amount of the martensitic phase of 17-4 ss could be increased if sintering was done in a nitrogen environment with MfMX.

### 3.2. Microstructure

#### 3.2.1. As Printed

The typical microstructure of the dense region of the AP Markforged-Metal-X-printed 17-4 ss is shown in Figure 6. The low magnification SEM images in Figure 6a,b exhibit the microstructures of the sample from two different perspectives. In Figure 6a, “V”-shaped ditches (or unfilled gaps) are seen at approximately equal intervals. The defects shown as straight lines and pits in these images were related to unfilled spaces between adjacent filaments. These voids were the spaces that were left unfilled due to the lack of insufficient melting of the stacked filaments through the thickness layers.

The images in Figure 7 were recorded in the regions between the structural defects (voids), showing some details of the microstructure. Figure 7a is an LOM image, which displays lath or plate-like microstructures, which is usually identified as a martensitic structure, while Figure 7b is an SEM image that shows different features of 17-4 AP. The non-lath/plate regions were expected to be in the austenite phase. This was consistent with the combined findings of the XRD and TEM that revealed the predominance of a martensite structure in AP. Although the cooling rate of the AM process was high enough to form a fully martensitic structure, some reverted austenite (inter-lath austenite) could be formed due to heat-affected regions occurring during the printing process [27].

On the other hand, Figure 7b shows the grains and grain boundaries clearly. The grains appear equiaxed with an average size of about 10 µm. The grain boundaries appear in a bright contrast compared with the core of the grains. Some nano-sized precipitates were formed along the grain boundaries, as indicated by white arrows in the image (Figure 7b). The precipitates were rich in Nb, as revealed by the EDS analysis. The dark contrast seen in both images is known to be pores of variable sizes.

Most of the larger pores occurred at the triple junctions and the edges were decorated with precipitates, as is clearly shown in Figure 8a, in which the variable-sized pores at/near the triple junctions are shown. Figure 8b is a typical EDS spectrum of a point analysis from the selected spots in Figure 8a. The list of Nb concentrations (wt. %) of the spots analyzed using EDS (Figure 8a) is given in Table 4. The amount of Nb in these precipitates thus lay between 24 and 31 wt. %. The phase of the precipitates is known to be Nb carbide (NbC) [18,27]. The diffusion of such heavier elements into the lattice defect, mainly to the grain boundaries, and its nucleation into carbides are the most likely phenomena during sintering. There can also be very fine Cu-rich precipitates in the matrix formed from the pre-existing Cu that is retained due to limited diffusion if the homogenization time is short [24]. Niobium carbide precipitates have an fcc structure and are incoherent with the bcc ferrite matrix that forms high-stress sites at the triple junctions. These precipitates are assumed to be responsible for the initiation and promotion of the pores and subsequent cracks at these sites. NbC precipitates are normally dissolved during a solid solution heat treatment, but the pores continue to exist.

#### 3.2.2. Post-Fabrication Heat Treatment

To obtain the optimal microstructure and mechanical strength of 17-4 ss, a two-step post-heat treatment (solid solution heat treatment and aging) is generally performed. The solid solution heat treatment of the current work was performed at 1038 °C, followed by precipitation hardening (aging) at 482 °C according to the H900 scheme. The LOM images of four samples (SA, H1A, H2A, H1Q) that exhibited the microstructure are shown in Figure 9. Irrespective of some differences in the heat treatment and cooling type, all the samples displayed an identical lath/plate-like microstructure that corresponded to martensite. The result was expected since the H900 heat treatment scheme is a well-established heat treatment scheme for obtaining the optimal mechanical strength following the formation of martensite structures and hardening precipitates. Although the samples were dominantly martensitic in the as-printed condition, some transformation of the fcc phase (γ phase) into the martensitic phase also occurred, as illustrated in Figure 4.

#### 3.2.3. Porosity

The porosity analysis was done without considering the pores/voids that lay between adjacent filaments, as shown in Figure 9b. A typical image of the pores analyzed in this study is given in Figure 10. The dark contrast/features seen in the back-scattered electron (BSE) image were generally pores/pits. The insert in Figure 10 is a secondary electron (SE) image that was intended to show pores that appear as a dark contrast in the BSE image. ImageJ (free software) was used to quantify the pores using several images of the as-printed samples recorded with SEM. The histogram shown in Figure 11 represents the average porosity of the individual surfaces/images. The sizes of the pores varied from a few nanometres to several micrometers. Most of the smaller pores were spherical, while the bigger ones were elongated along one axis. The average surface porosity was 3.47 ± 0.83%, which was very close to what was reported in the literature, for example, 3.3% in both [8] and [28]. Based on the porosity values obtained, the material density of the MfMX-17-4 ss from this study was approximately 96.6% disregarding the structural defects in connection with the voids observed in the adjacent regions of the filaments. This value was consistent with the relative density claimed by the Markforged company, i.e., ≥96%. However, in our assessment, the density we obtained in this work could not be taken as the material density of MfMX-17-4 ss because of the larger structural defects/voids that were not considered in the analysis.

Porosity is a common defect observed in AM-produced components that reduces the material density and consequently negatively affects the material strength. Metals with a high porosity density are weaker than those with a lower amount of porosity defects. As shown from the microscopy images, the Markforged 17-4 ss contains a high density of pores that were as large as several micrometers in size. In addition, the 17-4 ss component fabricated using Markforged Metal X was characterized by larger structural defects, including elongated pits/voids adjacent to deposited filaments. The structural defects (Figure 6) due to poor infusion of the filaments during printing were probably the main threats to the strength of the material. In general, the structural defects related to the porosity of MfMX-17-4 ss were significantly higher compared with materials fabricated using powder bed AM techniques. One aspect of the MfMX to be considered to reduce the porosity level and consequently increase the density of the material is the sintering temperature. Singh et al. [28], for instance, studied the relationship between porosity and sintering temperature and found a decreasing trend of the pores with increasing temperature. They observed that shrinkage of the pores with rising temperature in the range of 1200 to 1360 °C positively affected the density of MfMX-17-4 ss.

### 3.3. Hardness Testing

Hardness tests were performed on the surfaces parallel to the build direction. The average hardness calculated for all the samples tested are depicted in Figure 12. The deviations were quite large due to structural defects related to the porosity.

Referring to Figure 12, the highest and smallest average hardnesses measured were for H2A (417 ± 29 HV) and S1A (289 ± 36 HV), respectively. The other solid solution heat-treated sample (S1Q) also had a lower hardness than AP. As expected, the hardness of the samples that were only solid solution heat-treated, i.e., S1A and S1Q, became less hard than AP by approximately 16 and 11%, respectively. Before the solid solution heat treatment, the samples contained crystal defects, including grain boundaries and dislocation networks, which are high-stress locations. After the solid solution heat treatment, most of the dislocations were annihilated, and possibly, the grains were growing. In addition, some undesirable precipitates that formed during fabrication were dissolved and the phase of 17-4 ss became supersaturated with Cu [17,18]. The stress level was thus reduced and the material became weaker in strength.

However, all the samples that were aged (H1A, H1Q, H2A, H2Q, and DA) became harder than the as-printed samples. The samples that were solid solution heat-treated and aged were harder than AP by 17–24%. Similarly, the hardness of the aged-only (DA) sample was enhanced by about 23% compared with AP. This sample may have a similar microstructure as that of AP since the aging temperature (482 °C) was not high enough to release stresses in the 17-4 ss. At the aging temperature, nano-sized intermetallic precipitations are known to be formed from the super-saturated matrix. The interface between the nanoparticles and the matrix becomes a stress site. In addition, the precipitates prevent the movement of dislocations such that it resists deformation caused during indentation tests and consequently enhance the hardness/strength level of the material. The precipitates are mainly Cu-rich nanoparticles, as identified by TEM analysis in previous studies [17,18,19].

The list of some of the hardness measurements from previous reports is presented in Table 5. The average hardness of the MfMX-17-4 ss measured in this work was slightly larger than some of the hardnesses reported in the literature. As shown in Table 5, the average hardness of AP of this work was larger by 28 HV than the value reported in [16] and by 3 HV than the one in [24]. Similarly, the hardness of H900 (H2A) was larger by 63 HV than the one reported in [16] and by 42 HV than the one in [29]. On the other hand, the hardness values shown in [27] were larger than the corresponding values of both the AP and H900 samples in this work due to different measurement conditions. The load applied during the indentation for this work was 5 kg, whereas the load used by [27] was 0.2 kg. The superiority of the hardness from the current work relative to some values in the literature is not clear.

### 3.4. Tensile Testing

Typical engineering stress–strain curves of all the samples tested are shown in Figure 13. The numerical values of the yield strength (YS), ultimate tensile strength (UTS), and elongation (El) are also listed in Table 5, together with the results reported in the literature for comparative analysis. There were notable strength differences between the as-printed and the heat-treated samples. Generally, the UTS of the H900 samples was larger than 1000 MPa, whereas the UTS of AP was about 847 MPa. As shown in the stress–strain curve in Figure 13 and Table 5, the UTS of the heat-treated samples under the H900 condition was increased by 139–231 MPa. This was an increase of about 21–27% after the heat treatment due to the precipitation of hardening nanoparticles in the matrix during aging as discussed elsewhere in this paper. H1Q (solid solution heat-treated for 0.5 h/quenched and aged) was the sample that exhibited the largest YS (928 MPa) and UTS (1078 MPa), but a lower elongation (2.92%). In contrast, the smallest UTS (986 MPa) and elongation (2.52%) were measured for DA (only aged). The directly aged sample (no solid solution heat treatment) was strengthened only by the formation of precipitation hardening. Transformation of the austenite phase to the martensite phase was nil or insignificant at the aging temperature. This meant that the directly aged sample contained more austenite compared with those samples subjected to solid solution and aging heat treatments. When a considerable amount of the austenitic phase co-exists with the martensitic phase, the precipitation kinetics becomes sluggish because of the high solubility of copper in the austenite phase [7]. The relative concentration of hardening precipitates in the directly aged sample was thus lower than in the solution and aged samples. Consequently, the directly aged sample became less strong than those samples subjected to both heat treatments.

There were also some significant differences in the tensile properties due to different cooling rates (Figure 13 and Table 5). Under the same heat treatment conditions, the water-cooled samples exhibited better YS and UTS than the air-cooled samples (H1Q vs. H1A and H2Q vs. H2A). The exception was the UTS of H2A, which was slightly larger than that of H2Q. As expected, the air-cooled samples showed better ductility than the water-cooled (quenched) samples. This was consistent with the literature [19], which showed a higher tensile strength but lower elongation for samples treated under rapid cooling (water quenching) compared with samples cooled in the air [1].

The soaking time during the solid solution heat treatment affected the tensile properties, as shown in Table 5. For the air-cooled samples, those samples that were solid solution heat-treated for a longer soaking time, i.e., 1 h (H2A), showed better tensile strength (both YS and UTS) than those heat-treated for a shorter soaking time, i.e., 0.5 h (H1A), while the elongations were nearly the same. Contrary to this, higher strength and ductility were observed for the sample that was solid solution heat-treated for a shorter duration (H1Q) than the sample that was heat-treated for a longer soaking time (H2Q). The relative comparison of the tensile strength was also consistent with the hardness data. As shown in Table 5, H1Q was harder than H2Q by about 21 HV. The observation from this work was also in agreement with the reports in the literature [30]. The solid solution heat treatment scheme for the optimal mechanical strength for 17-4 ss manufactured using MfMX in this work was generally the one that combined a solid solution heat treatment for 0.5 h with cooling in water before aging. However, a more systematic investigation is required for confirmation and justification of these observations.

The tensile test of MfMX-17-4 ss of the current work showed comparable results with some other studies in the literature. For instance, the YS (551 MPa) and UTS (847 MPa) of AP in this work were very close to an L-PBF 17-4 ss [7], whose reported values for YS and UTS were 570 MPa and 944 MPa, respectively. Nevertheless, the tensile strength of 17-4 ss measured from the current work was generally lower than most of the observations in the literature (Table 5). A typical example is reported in [29], in which the YS and UTS of the L-PBF fabricated 17-4 ss exceeded the values obtained in the current work (MfMX-17-4 ss) by 110 MPa and 408 MPa, respectively. Similarly, the data reported on 17-4 ss manufactured using MfMX [28] and the values claimed by the Markforged company [16] were significantly larger than what was observed in this work. On the other hand, the study by Kedziora and co-workers [31] on the same material using the Markforged machine indicated a lower YS (441 MPa) and UTS (496 MPa) than even the current work. Although further study is required, such significant differences could have been due to the fabrication conditions. We presume that the materials in the literature might have been fabricated under optimal conditions (e.g., sintering temperature, soaking time, build orientation with reference to tensile load) relative to the current study, which utilized the default settings of the MfMX machine for printing.

The tensile strength measured in this work for 17-4 ss in the H900 condition was also lower than that of the conventional [7] and AM materials, as shown in Table 5. However, an optimal condition can be achieved from the MfMX system by adjusting different parameters, such as the sintering temperatures and orientations of the filaments. The ductility of the current work was also lower than the reports in the literature. The ductility of the material tested in this study was comparatively low (3.6% max vs. 11.7% in [29]). The brittle nature of the sample fabricated using the MfMX was primarily due to the orientation of the tensile test samples to the stress applied. The stress sites were mainly the voids, which were oriented perpendicular to the tensile direction, as pointed out by Kedziora et al. [31]. As shown in the next section, material failure occurred at the interfaces between the filaments, which is an indication of weak fusion between adjacent filaments. These form voids, especially when the distance between the layers/filaments is too large.

#### Fractography

The tensile fracture surfaces of all the tested samples were studied using SEM. The images that show the fracture behavior of the samples are presented in Figure 14. In all cases, the material fractures occurred in the gauge section of the samples. The grip sections of the tensile samples were excluded from the images shown in Figure 14. AP, H1A, and H2Q exhibited brittle failure characteristics, whereas DA, H2A, and H2Q appeared to have failed in a shear manner.

The characteristics of all the fracture surfaces demonstrated brittle-like fractures on the edges, as shown in the images in Figure 14. The edges of the test samples were the walls (Figure 2) where the long axis of the filaments was oriented parallel to the tensile loads. The middle region of the samples was made of “infills” whose filaments were oriented at ±45° with the axial orientation of the sample. However, the fracture surfaces of DA, H2Q, and H2A coincided with the interfaces between adjacent filaments. This indicated that the jointing between the filaments in the infill region was weaker compared with the walls. The load-carrying capacity of the filaments along the tensile axis was higher than when the filaments were deposited at an angle relative to the axis of the tensile test samples. A similar observation was reported by Todd et al. [8] for MfMX-fabricated 17-4 ss. According to this report [8], the ultimate tensile strength can be obtained when the filaments are aligned parallel with the loading direction.

Figure 15 presents details of the fracture surfaces whose images were acquired from the localities where stresses were supposed to concentrate during the tensile tests. The topographic images appear as intergranular brittle fractures with variable shapes, sizes, and distributions. A closer examination of the images reveals that the facets in the as-printed sample in Figure 15a were smaller than those of the rest of the samples, indicating a uniformly distributed granular structure. On the other hand, the microvoids in certain areas of the samples became coalesced, which is the characteristic of intergranular fracture. Some of these regions are shown with arrows in the images.

To understand the tensile behavior of materials, it is noteworthy to examine the details of the microstructures. The SEM images of the fracture surfaces shown in Figure 16a–d reveal incomplete jointing of the filaments due to a lack of fusion. The material failures were assumed to have been due to cracks that evolved due to the uniaxial stresses applied to the tensile samples. However, the cracks that occurred at the interfaces of the infill and wall (Figure 16c) were parallel to the applied stress. Similarly, the interfaces (voids) between adjacent filaments were parallel to the applied stress, as seen in the fracture surfaces of the samples. The latter indicated that the filaments were weakly joined to form compact/dense material. The wall was also weakly bonded with the infills, resulting in cracking when applying stresses during the tensile testing

In most cases, the cylindrical shapes of the filaments were maintained after sintering, which gave rise to a large volume fraction of unfilled spaces. Strong fusion occurred only at the tangential contacts of the filaments, while most of the curved regions remained free surfaces. This behavior is illustrated clearly in the image shown in Figure 16b. The fracture surfaces shown in the voids/gaps in the image exhibit the curved sections of the adjacent filaments that were not jointed due to lack of fusion. The voids appeared to be cracked, but they were rather unfilled spaces between non-joined parts of the filaments. Such structural defects reduced the load-carrying capacity of the material, which resulted in the brittle nature of the fractures.

## 4. Conclusions

In this work, the microstructure and mechanical properties of 17-4 PH stainless steel manufactured using a Markforged Metal X 3D printer in as-printed and heat-treated conditions were studied. The main observations and conclusions of the study were as listed below.

-The Markforged-Metal-X-manufactured 17-4 ss samples were predominantly martensitic in phase, with some retained austenitic phase. The XRD measurements exhibited partial transformation of the austenite phase into the martensite phase after the heat treatment. For the complete transformation of the austenite phase into an α′-bcc phase, a different heat treatment scheme than the H900 needs to be considered.-The microstructure of the Markforged-Metal-X-manufactured 17.4 ss was characterized by patterned voids or unfilled spaces that could eventually affect the mechanical integrity of the material. The voids are believed to have occurred mainly due to a lack of fusion of adjacent filaments.-The porosity analyzed in the dense region of the material was about 3.5%. Most of the larger pores observed were at the triple junctions that were rich in NbC precipitates.-Post-fabrication heat treatment enhanced the hardness by 17–28% and the tensile strength by 21–27%, depending on the heat treatment conditions.-The hardnesses and the tensile strengths of the samples under directly aged conditions were slightly lower than those samples tested after the solid solution and precipitation hardening heat treatment. The difference could have been due to the transformation of austenite into martensite structures following the solid solution heat treatment.-The hardness of the as-printed samples (331 ± 28 HV) and heat-treated samples under the H900 condition (417 ± 29 HV) were comparable with the reported values in the literature.-The maximum elongation achieved was about 3.6%, which was comparatively low relative to results from other AM techniques.-The maximum ultimate tensile strengths from the as-printed and H900 conditions were 551 and 1078 MPa, respectively. These were lower than most of the observations for conventional and AM methods.-The presumption for lower-strength 17-4 ss from the current study relative to reports in the literature based on the Markforged Metal X printer may have been due to differences in fabrication conditions (e.g., sintering temperature, soaking time, and build orientation concerning tensile load).-The inferiority of the tensile strength relative to other AM techniques could be related to incomplete fusion of the filament; this might have been due to the low sintering temperature, which left behind pattered voids; the infill orientation relative to the applied load; and the high pore concentration.-The results indicated that the Markforged printer is a promising technology given that the printing processes are fully developed and optimized.

## Figures and Tables

**Figure 1 materials-15-06898-f001:**
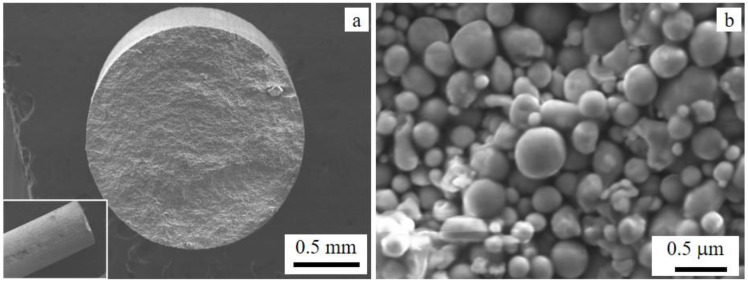
SEM images of a 17-4 ss filament (feedstock material): (**a**) cross-sectional view of a filament and (**b**) high-resolution image showing components of the filament. The inset in (**a**) is the axial view of the filament at low magnification.

**Figure 2 materials-15-06898-f002:**
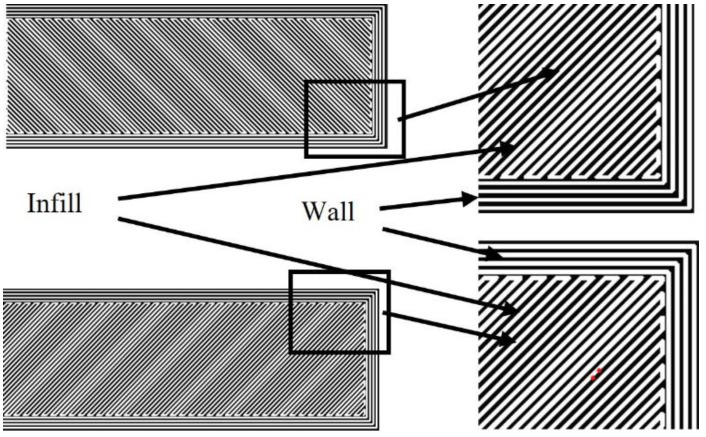
Orientation of filaments in the successive layers of parts.

**Figure 3 materials-15-06898-f003:**
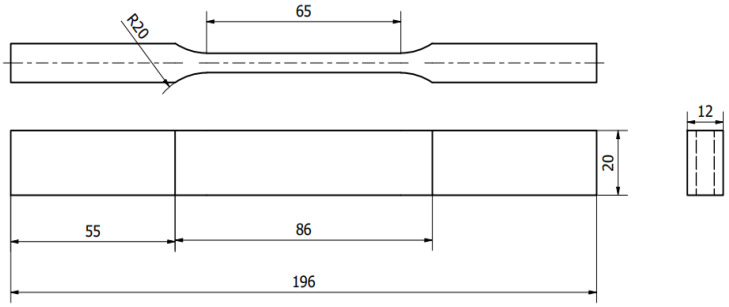
Geometry and dimension of the tensile test specimen.

**Figure 4 materials-15-06898-f004:**
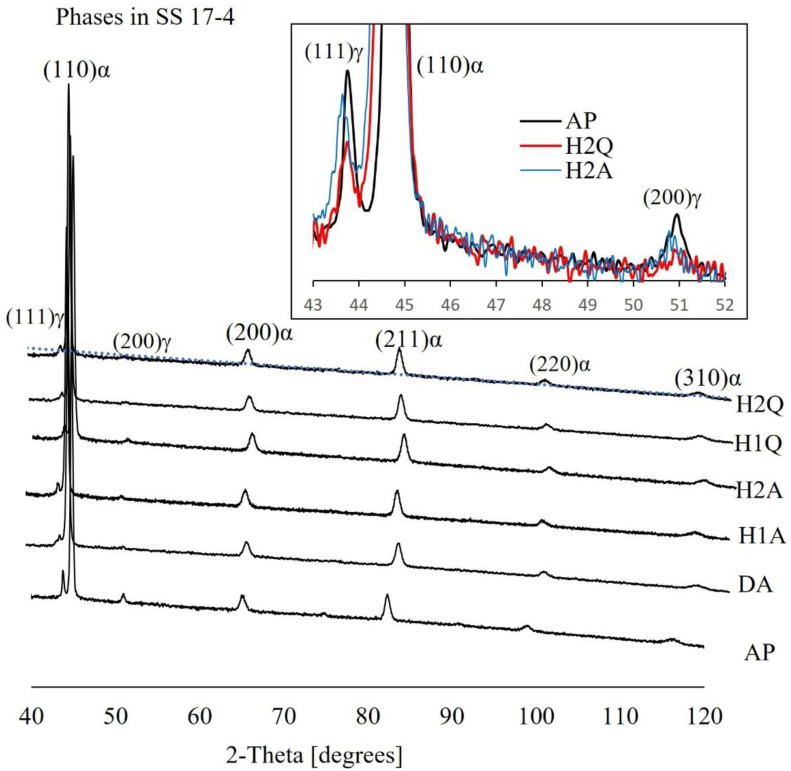
XRD diffractograms of the MfMX-17-4 ss samples before and after the heat treatment. The insert is the magnified view of the low-index peaks of the fcc crystal planes.

**Figure 5 materials-15-06898-f005:**
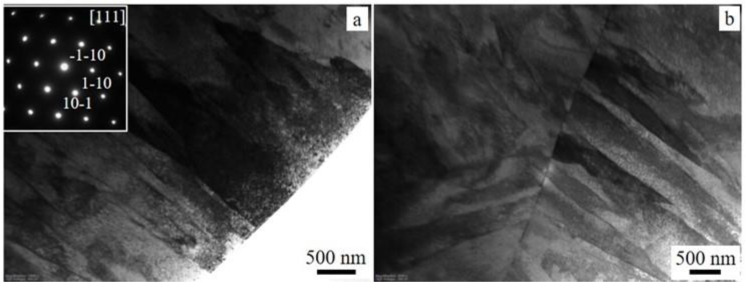
TEM bright-field images of the as-printed 17-4 ss (**a**,**b**). The insert in (**a**) is a selected area diffraction pattern recorded along the [111] zone axis from the image. The image in (**b**) shows the grain boundary and lath structures.

**Figure 6 materials-15-06898-f006:**
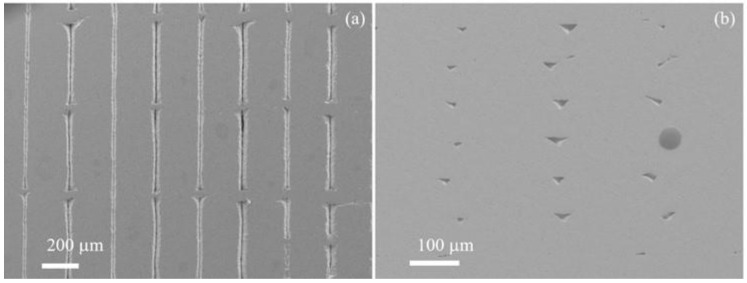
(**a**,**b**) SEM images of the as-printed sample surface structures from two perspectives.

**Figure 7 materials-15-06898-f007:**
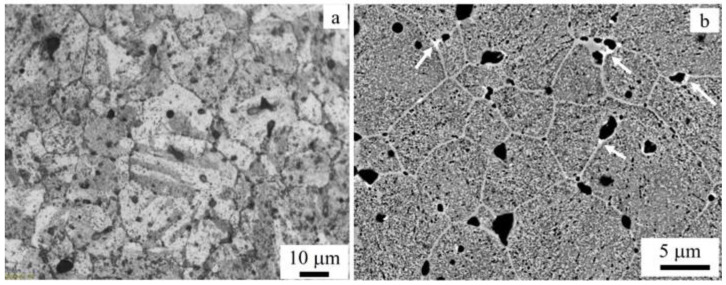
The microstructure of MfMX-17-4 AP stainless steel: (**a**) LOM and (**b**) SEM (back-scattered electron image).

**Figure 8 materials-15-06898-f008:**
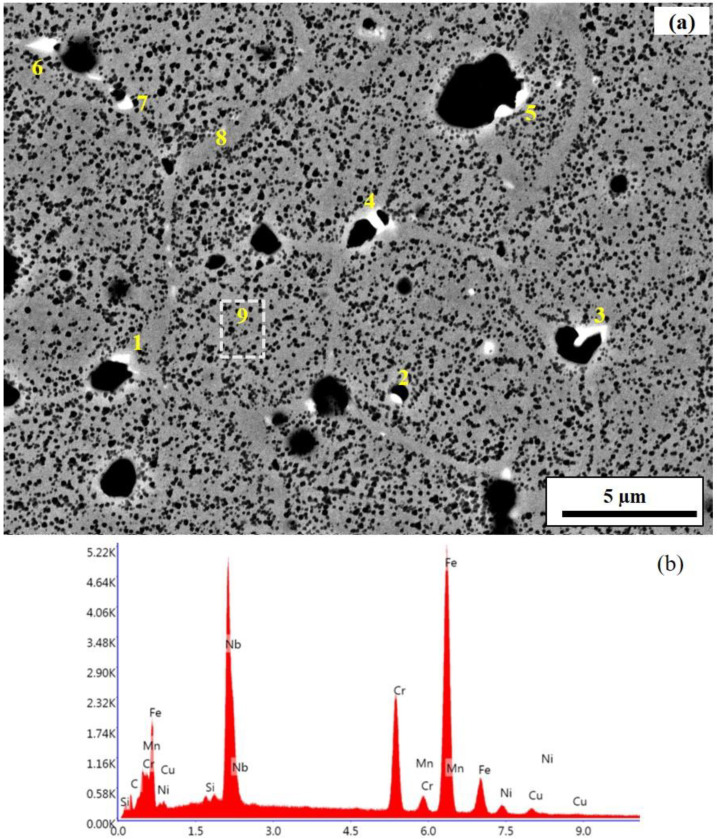
EDS elemental composition analysis of Nb-rich precipitates in AP: (**a**) back-scattered SEM image and (**b**) typical EDS spectrum. The horizontal axis of the EDS spectrum is the energy in keV, whereas the vertical axis is the number of X-ray counts or intensity. (Note: The “K” in the vertical axis is equivalent to 1000, i.e., 1000 is multiplied to the numbers in the vertical axis).

**Figure 9 materials-15-06898-f009:**
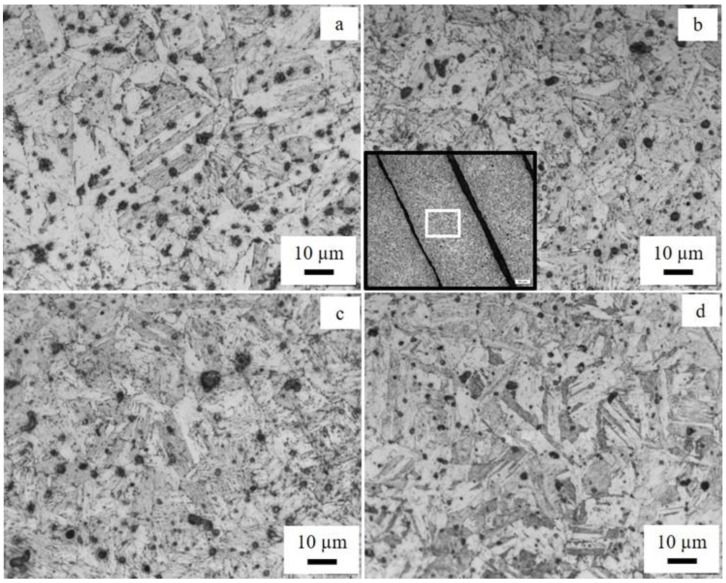
LOM images of heat-treated samples: (**a**) SA, (**b**) H1A, (**c**) H2A, and (**d**) H1Q. The inset in Figure 9b is a small-magnification image showing regions where the magnified images were recorded. The voids between the filaments appear as a dark contrast. The white square in the inset is the approximate location from where the magnified image of Figure 9b was acquired.

**Figure 10 materials-15-06898-f010:**
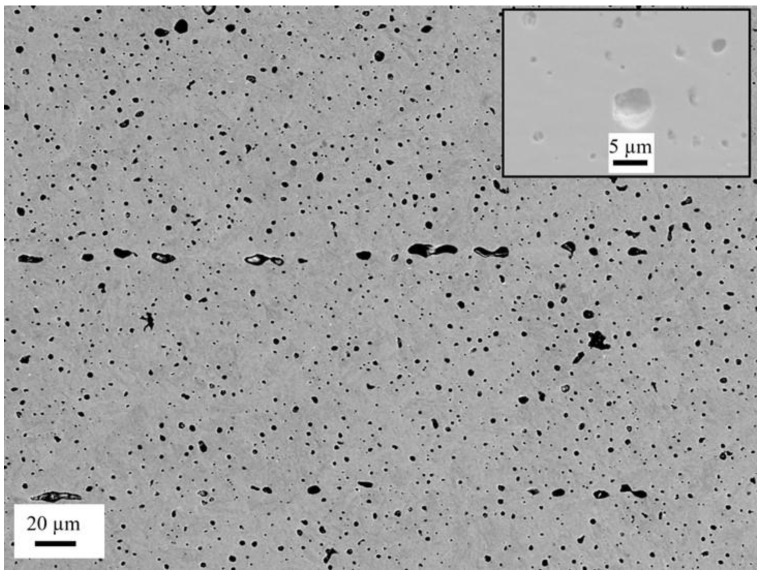
An example of a back-scattered electron image showing pores with variable sizes. The insert is a higher magnification secondary electron image to illustrate the pore structures.

**Figure 11 materials-15-06898-f011:**
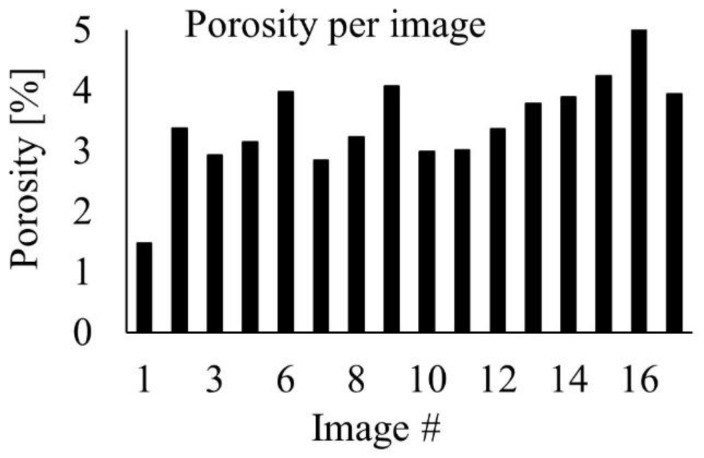
Histogram showing the average porosity of the individual images.

**Figure 12 materials-15-06898-f012:**
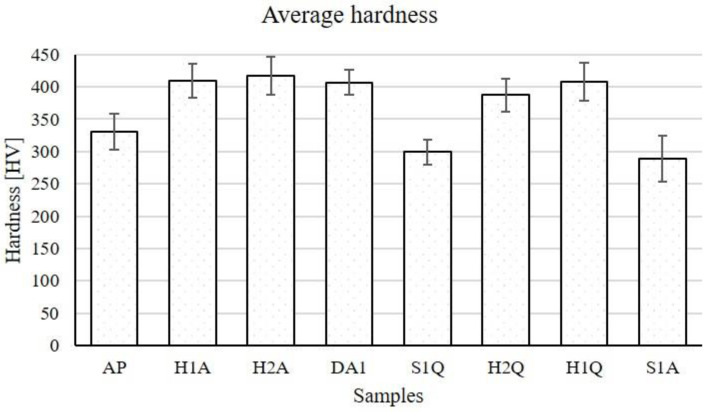
The average hardness of all specimens tested.

**Figure 13 materials-15-06898-f013:**
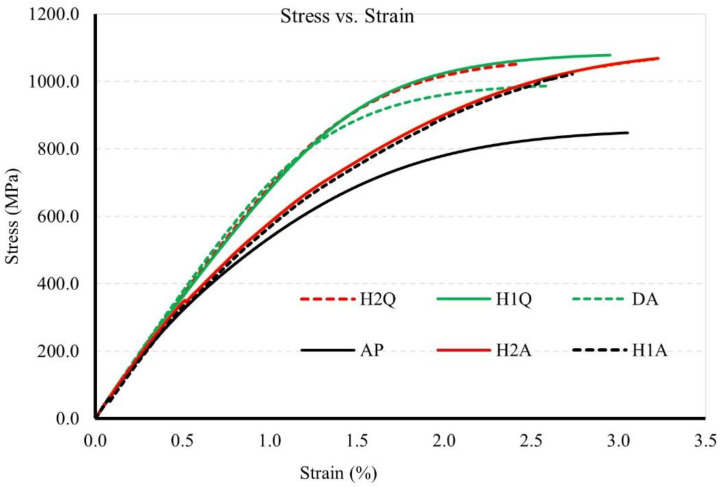
Engineering stress–strain curves of the as-printed and heat-treated samples.

**Figure 14 materials-15-06898-f014:**
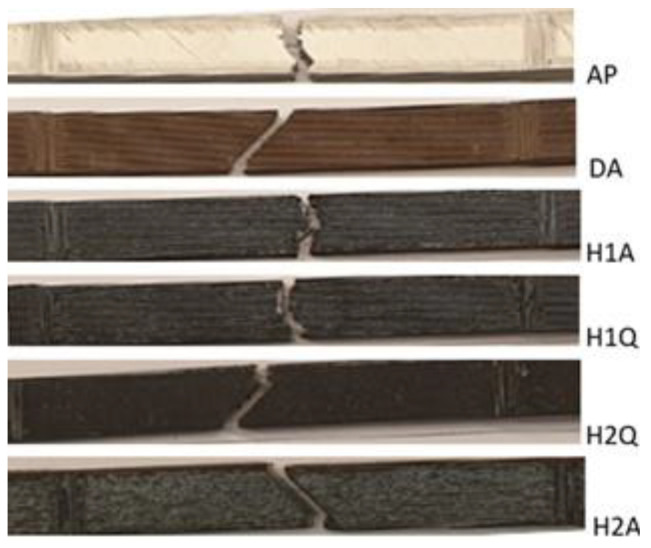
Fracture profile of material failure during the tensile test.

**Figure 15 materials-15-06898-f015:**
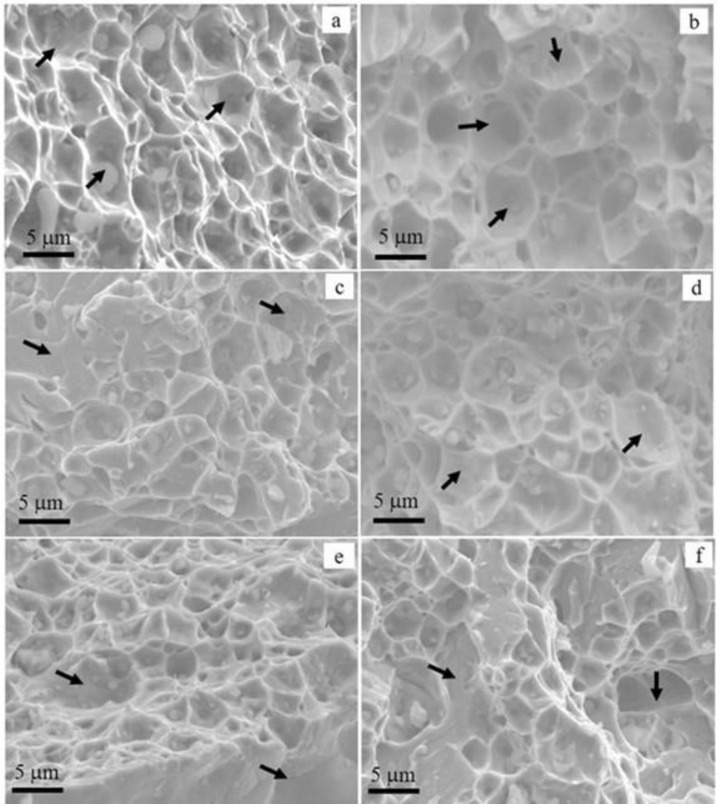
SEM image showing the fracture morphology of (**a**) AP (**b**) DA, (**c**) H1A, (**d**) H2A, (**e**) H1Q, and (**f**) H2Q.

**Figure 16 materials-15-06898-f016:**
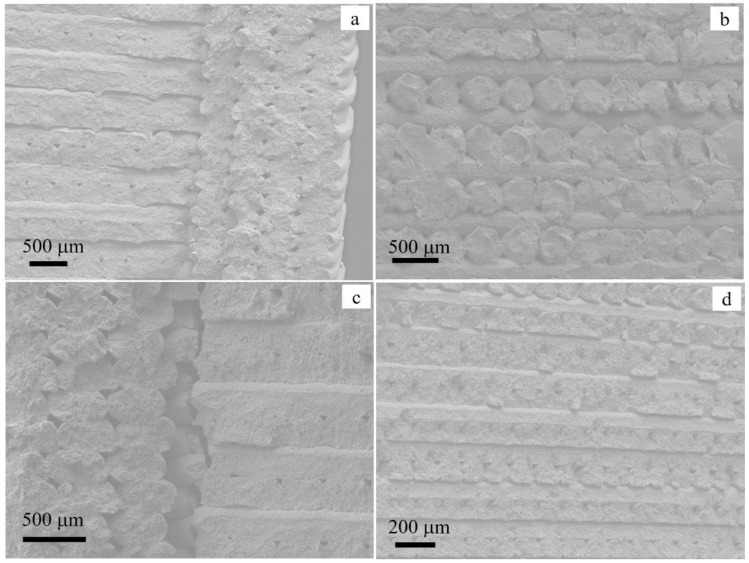
SEM images of the tensile fracture surface with different filament orientations: (**a**) infill on the left and wall on the right/edge, (**b**) alternating filaments in the infill, (**c**) image showing cracks at the interface of infill and wall, and (**d**) filaments oriented in the same direction.

**Table 1 materials-15-06898-t001:** Composition of the 17-4 ss powder/filament.

Composition	Cr	Ni	Cu	Si	Mn	Nb	C	P	S	Fe
wt.%	15–17	3–5	3–5	1 max	1 max	0.15–0.45	0.07 max	0.04 max	0.03 max	Bal

**Table 2 materials-15-06898-t002:** Printing parameter.

Orientation of Part	(X, Y, Z) (180°, 0°, 180°)
Layer thickness	0.125 mm after sintering
Infill	Solid infill +45/−45° orientation
Wall	4 walls
Surface boundary	1
Sintering T	Ca. 85% of melting T
Printing chamber T	45 °C

**Table 3 materials-15-06898-t003:** Descriptions of heat treatment scheme.

Sample	Solid Solution Heat Treatment (SHT)		Aging		Cooling
	T (°C)	Soaking Time (h)	T (°C)	Time (h)	
H1A	1038	0.5	482	1	Air
H2A	1038	1	482	1	Air
H1Q	1038	0.5	482	1	Water
H2Q	1038	1	482	1	Water
DA	----	----	482	1	Air
S1A	1038	0.5	….	….	Air
S1Q	1038	0.5	….	….	Water
AP	As printed				

**Table 4 materials-15-06898-t004:** The concentration of Nb at the spots shown in Figure 8a.

Spot	1	2	3	4	5	6	7	8	9
wt. %	26	24	36	25	30	31	28	0	0

**Table 5 materials-15-06898-t005:** List of tensile strengths and hardnesses compared with the literature.

Sample Condition	AM	SHT	Aging	YS	UTS	El	Hardness	Reference
		°C/h	°C/h	MPa	MPa	%	HV	
AP	MfMX			551	847	3.1	331 ± 28	This work
SA	MfMX		482	832	986	2.6	407 ± 20	This work
H1A	MfMX	1038	482	513	1021	3.6	409 ± 26	This work
H2A	MfMX	1038	482	602	1075	3.4	417 ± 29	This work
H1Q	MfMX	1038	482	921	1078	3.0	408 ± 29	This work
H2Q	MfMX	1038	482	892	1051	2.4	387 ± 26	This work
AP	MfMX			800	1050	5	302	[16]
H900	MfMX	1038	482	1100	1250	6	354	[16]
AP	MfMX			823	940	3.67		[28]
AP	L-PBF			661 ± 24	1255 ± 3	9.9 ± 0.2	333 ± 2	[29]
H900	L-PBF		480/1	945 ± 12	1417 ± 6	11.7 ± 0.8	375 ± 3	[29]
AP	L-PBF						334.5 ± 15	[27]
H900	L-PBF	1038/4	482/1				524.5 ± 6	[27]
As built	Wrought						384.3 ± 8	[27]
H900	Wrought	1038/4	482/1				450.1 ± 9	[27]
AP	L-PBF			784	922	16.7	328	[24]
H900	L-PBF	1038/1/AC	482/1	1280	1399	10.5		[24]
AP	L-PBF			570	944			[7]
	L-PBF	788/2	482/1	1126	1457			[7]

## Data Availability

Not applicable.

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
