# Peer review of "Investigations of the Microstructure and Mechanical Properties of 17-4 PH ss Printed Using a MarkForged Metal X"

_materials, 2022, doi:10.3390/ma15196898_

Round 1
Reviewer 1 Report
1. Please explain the parameter design of solution temperature of 1038 °C in this study.
2. Authors indicated that the samples were cooled in air or in water (quenched) after the soaking. Please provide the microstructure characteristics of the sample cooled in air and in water to clarify the relationship between microstructures and residual stresses.
3. Please explain the purpose of aging temperature of “482 °C”.
4. Low angle (10~30 degree) of the XRD pattern of Fig.4 should be provided. In addition, the discussion of the effects of heat treatment parameters (AP, DA, H1A, H2A, H1Q, H2Q) on the angle of alpha phases in the XRD pattern is necessary.
5. In page 7, what does SLM-fabricated ss 17-4 mean….?
6. Authors indicated that the cooling rate of Am process is high enough to form fully martensite structure, and some reverted austenite can be formed (XRD and TEM results). Please provide the ratio of martensite and austenite structures.
7. In Fig. 7, what composition are the black particles (or pores?) in MfMX-17-4 AP stainless steel.
8. 17-4 is a stainless steel material that can be strengthened by precipitation hardening. However, the relevant precipitates were not found in the aging samples. In addition, the aging sample should be conducted to TEM analysis to confirm the characteristics of the precipitates.
9. The manuscript requires a better improvement in terms of language and grammar usage.
1. Please explain the parameter design of solution temperature of 1038 °C in this study.
2. Authors indicated that the samples were cooled in air or in water (quenched) after the soaking. Please provide the microstructure characteristics of the sample cooled in air and in water to clarify the relationship between microstructures and residual stresses.
3. Please explain the purpose of aging temperature of “482 °C”.
4. Low angle (10~30 degree) of the XRD pattern of Fig.4 should be provided. In addition, the discussion of the effects of heat treatment parameters (AP, DA, H1A, H2A, H1Q, H2Q) on the angle of alpha phases in the XRD pattern is necessary.
5. In page 7, what does SLM-fabricated ss 17-4 mean….?
6. Authors indicated that the cooling rate of Am process is high enough to form fully martensite structure, and some reverted austenite can be formed (XRD and TEM results). Please provide the ratio of martensite and austenite structures.
7. In Fig. 7, what composition are the black particles (or pores?) in MfMX-17-4 AP stainless steel.
8. 17-4 is a stainless steel material that can be strengthened by precipitation hardening. However, the relevant precipitates were not found in the aging samples. In addition, the aging sample should be conducted to TEM analysis to confirm the characteristics of the precipitates.
9. The manuscript requires a better improvement in terms of language and grammar usage.

Author Response
Response – Reviewer #1
We would like to thank the reviewer for reading the manuscript critically and providing useful comments and suggestions. Corrections/responses to the comments of the reviewer are given below. The changes/updates are highlighted in yellow colour in the manuscript. The changes related to language/grammar are shown in blue fonts.
- Please explain the parameter design of solution temperature of 1038 °C in this study.
Response: We have adopted the H900 heat treatment scheme for attaining optimum conditions (mechanical properties) as indicated in the introduction section. H900 is a well-known post-heat treatment procedure for the conventionally produced 17-4 SS in industries and among scholars. We have adopted this scheme since we didn’t find other options yet that is taken as standard for the AM material heat treatment scheme. Our observation however indicates that the H900 scheme transforms only a fraction of the retained austenite phase into martensite. The reviewer’s comment thus reminds us of a different heat treatment scheme other than the H900 for AM-fabricated material. So, we have included the following statement in our conclusion that motivates us/others to design a different post-heat treatment for 17-4 SS fabricated by the MfMX machine.
‘For the complete transformation of austenite into α¢-bcc, a different heat treatment scheme than the H900 needs to be considered.’
- Authors indicated that the samples were cooled in air or water (quenched) after the soaking. Please provide the microstructure characteristics of the sample cooled in air and in water to clarify the relationship between microstructures and residual stresses.
Response: We have raised it under subsection 3.2.2 and presented OM images for different cases (AP, air-cooled, and water-cooled). Inspection of several of these images shows quite a similar microstructure characterized by the lath-type martensite and we have presented some of the images from the analysis. This is also supported by XRD results presented in Section 2.4 (Figure 4). After the heat treatment, as shown in Fig. 4, the retained austenite phase has been partially transformed into the alpha bcc phase (martensite). It can then be assumed that the number of martensitic laths/plates has increased after heat treatment. But it is difficult to notice the changes between the different samples (especially air-cooled vs. water-cooled). This can however be noticeable by closely examining the XRD peaks. The inset image (magnified) in Fig. 4, compared to the low index peaks (110 for bcc and 111, 200 for austenite) of AP (as printed) vs. H2A (air-cooled), and H2Q (water-cooled). From this comparison, one can visualize that the heat treatment results in peak broadening, which show an indication of an increased stress level. It is rational to say here that the broadening might be caused mainly due to the formation of the tiny hardening precipitates after the heat treatment. Presumably, this is a fact, but we did not want to discuss it in detail since investigations related to stress were not our primary objective in this study. But to meet the reviewer’s comment, we have modified the inset in Fig. 4 to illustrate the changes after heat treatment for both air-cooled and water-cooled samples to visualize the extent of broadening and thus the stresses relative to the as printed sample.
- Please explain the purpose of aging temperature of “482 °C”.
Response: This is discussed in the Introduction section in the 4th paragraph. In addition, the sentence in Section 2.2 is modified as follows: ’To increase the mechanical strength through the formation of precipitation hardening, the samples were aged at 482 °C for 1 h, each.’ As shown in the literature [24], Cu-rich fine particles which are coherent with the bcc-matrix are precipitated after the aging heat treatment.
- Low angle (10~30 degrees) of the XRD pattern of Fig.4 should be provided. In addition, the discussion of the effects of heat treatment parameters (AP, DA, H1A, H2A, H1Q, H2Q) on the angle of alpha phases in the XRD pattern is necessary.
Response: The two knowns (identified) phases are having their lowest hkl-index (111 for fcc and 110 for bcc) peaks above 35 degrees (2-Theta). If so, why do we need to have or map the low angle (10~30 degrees) 2-Theta in this material (17-4 ss)? We have however compared and discussed the effects of heat treatment relative to the hkl planes (111 & 200 for fcc and 110 for bcc) in the low angle range under Section 3.1.
- In page 7, what does SLM-fabricated ss 17-4 mean….?
Response: We have now stated the acronym for the ‘SLM’ on page 7, as Selective Laser Melting (SLM) and modified the expression as ‘Selective Laser Melting (SLM)-fabricated ss 17-4’.
- Authors indicated that the cooling rate of Am process is high enough to form a fully martensite structure, and some reverted austenite can be formed (XRD and TEM results). Please provide the ratio of martensite and austenite structures.
Response: One way of exploring such data is from the XRD diffractogram, using the whole-pattern or single-peak analysis. As shown in Fig. 4, the peaks that belong to the austenite phase are too weak to give a reliable measurement. So, we prefer to analyze the data qualitatively than quantitatively.
- In Fig. 7, what composition are the black particles (or pores?) in MfMX-17-4 AP stainless steel?
Response: The black contrast/particles are pores as indicated in the manuscript. However, Nb-based precipitates are observed at the edges of the pores as presented in sub-section 3.2.1 (Fig. 8). Nb-precipitates are considered (especially at the triple junction) as one of the causes for the initiation and propagation of the pores.
- 17-4 is a stainless steel material that can be strengthened by precipitation hardening. However, the relevant precipitates were not found in the aging samples. In addition, the aging sample should be conducted to TEM analysis to confirm the characteristics of the precipitates.
Response: The intention of aging this material as many of the previous studies show is to form very fine Cu-based hardening precipitates. They are generally less than a few nm in size. Observation of these precipitates requires advanced TEM (aberration-corrected, HAADF/BF STEM, FEG) for reliable identification. The TEM we have is an LB6-gun with an ordinary STEM system of BF and DF detectors. They may be observed by the diffraction approach, but it requires an intensive job (very good sample, spending a lot of time on the microscope – laborious). In addition, since this study is not funded (not project-based), we couldn’t afford the cost of characterizing the samples using advanced techniques locally or abroad.
- The manuscript requires a better improvement in terms of language and grammar usage.
Response: The manuscript is revised by the authors and the observed technical and grammatical errors are corrected. The changes are shown in blue color in the manuscript.

Reviewer 2 Report
The authors have done an interesting work on design and investigating microstructure and mechanical properties of MarkForged metal x printed part. This new technology that allows to obtain metallic components by a post-printing sintering process is very interesting for its low cost. The results of this manuscript are very interesting for future developments and specific applications. The paper needs some changes to be considered for publication:
1. Additive manufacturing allows you to create components characterized by extremely particular shapes and precisely defined by CAD software, for these reasons you can cite the papers:
- Kedziora, S.; Decker, T.; Museyibov, E.; Morbach, J.; Hohmann, S.; Huwer, A.; Wahl, M. Strength Properties of 316L and 17-4 PH Stainless Steel Produced with Additive Manufacturing. Materials 2022, 15, 6278. https://doi.org/10.3390/ma15186278
- Almonti, D., & Ucciardello, N. (2019). Design and thermal comparison of random structures realized by indirect additive manufacturing. Materials, 12(14), 2261. https://doi.org/10.3390/ma12142261
- Saeed, K.; McIlhagger, A.; Harkin-Jones, E.; McGarrigle, C.; Dixon, D.; Archer, E. Elastic Modulus and Flatwise (Through-Thickness) Tensile Strength of Continuous Carbon Fibre Reinforced 3D Printed Polymer Composites. Materials 2022, 15, 1002. https://doi.org/10.3390/ma15031002
- Almonti, D., Baiocco, G., Mingione, E., & Ucciardello, N. (2022). FEM Simulations for the Optimization of the Inlet Gate System in Rapid Investment Casting Process for the Realization of Heat Exchangers. International Journal of Metalcasting, 16(3), 1152-1163. https://doi.org/10.1007/s40962-021-00668-7
2. Figure 2 (b) is blurred, the authors may add arrows to make the reader easier;
3. Table 2 shows the parameters used to make the samples. Authors should argue why these values were chosen, if they are default parameters provided by the software or literature;
4. In 2.2 Heat treatment section, the authors should indicate the furnace model used in the experimentation;
5. In 2.2 Heat treatment section, “To avoid undesirable phase transformations at the lower temperatures, the samples were introduced after stabilizing the furnace to the given target temperature”, the authors should better argue if this operation does not significantly change the temperature reached inside the oven;
6. Have the authors replicated the samples in order to observe a repeatability of characteristics obtained under the same treatment conditions? This should be better argued;
7. Table 3 is broken on two pages, authors should correct this;
8. In 2.3. Hardness and tensile testing section, “The interval between adjacent indentations was 1 mm and the closest indentation near the edge was about 3 mm”, What is the wall thickness measurement? in the text the authors do not say the diameter of the extruder of the printer and if they measured the size of the wall so as to understand if hardness tests may have been carried out on the wall or on the infill and if this resulted in a different mechanical behavior. Authors should better argue this aspect;
9. Figure 7 has the caption detached from the figure, the authors should correct this;
10.In Figure 13 the trace relative to H2A is not distinguished, the authors should modify the graph so as to render visible all the results;
Author Response
Response – Reviewer #2
We would like to thank the reviewer for reading the manuscript critically and providing useful comments and suggestions. Corrections/responses to the comments of the reviewer are given below. The changes in the manuscript are highlighted in green color. Changes related to language improvement are shown in blue.
- Additive manufacturing allows you to create components characterized by extremely particular shapes and precisely defined by CAD software, for these reasons you can cite the papers:
- Kedziora, S.; Decker, T.; Museyibov, E.; Morbach, J.; Hohmann, S.; Huwer, A.; Wahl, M. Strength Properties of 316L and 17-4 PH Stainless Steel Produced with Additive Manufacturing. Materials 2022, 15, 6278. https://doi.org/10.3390/ma15186278
- Almonti, D., & Ucciardello, N. (2019). Design and thermal comparison of random structures realized by indirect additive manufacturing. Materials, 12(14), 2261. https://doi.org/10.3390/ma12142261
- Saeed, K.; McIlhagger, A.; Harkin-Jones, E.; McGarrigle, C.; Dixon, D.; Archer, E. Elastic Modulus and Flatwise (Through-Thickness) Tensile Strength of Continuous Carbon Fibre Reinforced 3D Printed Polymer Composites. Materials 2022, 15, 1002. https://doi.org/10.3390/ma15031002
- Almonti, D., Baiocco, G., Mingione, E., & Ucciardello, N. (2022). FEM Simulations for the Optimization of the Inlet Gate System in Rapid Investment Casting Process for the Realization of Heat Exchangers. International Journal of Metalcasting, 16(3), 1152-1163. https://doi.org/10.1007/s40962-021-00668-7
Response: From the suggested references, we found only the first reference which has direct relevance to the current study. Reference #1 is now cited under Section 3.4.
- Figure 2 (b) is blurred, the authors may add arrows to make the reader easier;
Response: The figure is improved and replaced the former one in the manuscript.
- Table 2 shows the parameters used to make the samples. Authors should argue why these values were chosen if they are default parameters provided by the software or literature;
Response: The parameters are default, and they were provided by Eiger software which is used by the Markforged Metal X 3D metal printer.
- In 2.2 Heat treatment section, the authors should indicate the furnace model used in the experimentation;
Response: The model of the furnace is Nabertherm P300, Lilienthal/Bremen, Germany. It is added in the manuscript.
- In 2.2 Heat treatment section, “To avoid undesirable phase transformations at the lower temperatures, the samples were introduced after stabilizing the furnace to the given target temperature”, the authors should better argue if this operation does not significantly change the temperature reached inside the oven;
Response: To minimize the variation of the furnace temperature, this operation has been done by two persons. One person is opening the furnace door while the other person, who is readily holding the sample(s) quickly puts it into the furnace. Then, the other person closed the door instantly. The operation generally took only a few seconds. In doing so, the temperature only drops by less than 10 °C. As we have experienced the temperature reaches again a target value in a matter of few seconds. The temperature variation due to this operation is so small and the resulting furnace temperature is by far higher than the temperature region that causes phase changes.
- Have the authors replicated the samples in order to observe a repeatability of characteristics obtained under the same treatment conditions? This should be better argued;
Response: For every test, six specimens were fabricated by Markforged metal x 3D printer under the same parameter conditions (Table 2) as described in the standard used to observe the repeatability. The analysis was based on the average result for every test. A sentence is now added in Section 2.3 in response to the comment.
‘ To ensure production repeatability, six samples were fabricated under the same parameters (Table 2) for every condition shown in Table 3 for the tensile measurements.’
- Table 3 is broken on two pages, authors should correct this;
Response: It is corrected now.
- In 2.3. Hardness and tensile testing section, “The interval between adjacent indentations was 1 mm and the closest indentation near the edge was about 3 mm”, What is the wall thickness measurement? in the text the authors do not say the diameter of the extruder of the printer and if they measured the size of the wall so as to understand if hardness tests may have been carried out on the wall or on the infill and if this resulted in a different mechanical behavior. Authors should better argue this aspect.
Response: The diameter of the filament for both wall and infill is the same, which is 0.125mm.
- Figure 7 has the caption detached from the figure, the authors should correct this;
Response: Corrected.
- In Figure 13 the trace relative to H2A is not distinguished, the authors should modify the graph so as to render visible all the results;
Response: Figure 13 is now corrected.

Reviewer 3 Report
The authors have performed fascinating research for the use of manufacturing industries. Some queries need to be addressed before accepting the manuscript for possible publication.
1. The authors mentioned that very few research papers had addressed the use of the “Markforged metal X (MfMX) printing machine” is there any specific reason why only a few researchers have used this type of AM? In the Introduction, part authors mentioned “is less complex, inexpensive and with larger printing volume capacity,” then why not the researchers perform this type of AM method to solve complex manufacturing problems?
2. Do mention the type of filament used in the present work.
3. What is the melting temperature used in the present work?
4. Authors must mention the ASTM standards used to carry out the mechanical testing.
5. Results and discussions are well explored with the surface analysis techniques.
Author Response
Response - Reviewer #3
We would like to thank the reviewer for reading the manuscript critically and providing useful comments and suggestions. Corrections/responses to the comments of the reviewer are given below. The changes are highlighted in red color in the manuscript. Changes related to language improvement are shown in blue.
- The authors mentioned that very few research papers had addressed the use of the “Markforged metal X (MfMX) printing machine” is there any specific reason why only a few researchers have used this type of AM? In the Introduction, part authors mentioned “is less complex, inexpensive and with larger printing volume capacity,” then why not the researchers perform this type of AM method to solve complex manufacturing problems?
Response: As mentioned in the ‘Introduction’ section of the manuscript, the machine is relatively new on the market, and we have assumed that the machine has been acquired only by a few users. We hope that many more reports will come out sooner or later based on the Markforged method.
- Do mention the type of filament used in the present work.
Response: The filament used is a wire-type with a diameter of 1.75 mm 17-4 stainless steel mixed with wax. It is shown in Figure 1 and described in Section 2.1.
- What is the melting temperature used in the present work?
Response: The process involved is not melting but sintering. The sintering temperature adopted by the manufacturer of the Markforged Metal X is about 85% of the melting temperature of 17-4 SS.
- Authors must mention the ASTM standards used to carry out the mechanical testing.
Response: As mentioned on page 5 (Section 2.3), for the tensile test, we used ISO standards and not ASTM. The standard used for the tensile test was ISO6892-1.
- Results and discussions are well explored with the surface analysis techniques.
Response: The objectives of the study as indicated in the paper are to characterize and analyse the microstructure and mechanical properties of 17-4 ss manufactured by the Markforged AM technique before and after the standard heat treatment scheme. Although surface characterization such as the extent of roughness is an important issue related to AM-fabricated materials, our focus was on the bulk analysis. So, we didn’t employ surface analysis techniques and we do not have data on the surface analysis to present and discuss the results. But we appreciate very much the reviewer’s comment about the topic, and we shall consider carrying it out in our upcoming investigations on the same material.

Round 2
Reviewer 2 Report
The authors have corrected the manuscript and it can now be published